# Pharmacokinetic profiles of Moutan Cortex after single and repeated administration in a dinitrobenzene sulfonic acid-induced colitis model

Jin-Hwa Kim[1], Ji-Soo Jeong[1], Jeong-Won Kim[1,2], Eun-Hye Chung[1], Su-Ha Lee[1], Je-Won Ko[1], Youn-Hwan Hwang[3]*, Tae-Won Kim[1]*

1 College of Veterinary Medicine (BK21 FOUR Program), Chungnam National University, Daehak-ro, Daejeon, Republic of Korea, 2 Laboratory of Radiation Exposure and Therapeutics, National Radiation Emergency Medical Center, Korea Institute of Radiological and Medical Sciences, Seoul, Republic of Korea, 3 Herbal Medicine Research Division, Korea Institute of Oriental Medicine, Daejeon, Republic of Korea

* hyhhwang@kiom.re.kr (HYH); taewonkim@cnu.ac.kr (TWK)

## Abstract

Moutan Cortex (MC), the dried root bark of *Paeonia suffruticosa*, is used in traditional Chinese and Korean medicine to treat enteritis for its anti-inflammatory properties. This study compared the pharmacokinetic (PK) profiles of paeonol and paeoniflorin in normal and dinitrobenzene sulfonic acid (DNBS)-induced colitis rats, and to determine how repeated low-dose MC [MC(L), 0.5 g/kg] or high-dose MC [MC(H), 2.5 g/kg] alters PK and disease severity. Using ultra-performance liquid chromatography–tandem mass spectrometry, we found that DNBS modestly increased paeonol $AUC_{last}$ (NC: $247.8 \pm 63.7$ vs DNBS: $337.0 \pm 120.8$ hr*ng/mL) and decreased paeoniflorin (NC: $474.1 \pm 11.7$ vs DNBS: $463.7 \pm 106.8$ hr*ng/mL) compared to controls (ns). After repeated dosing, the maximum plasma concentration ($C_{max}$) of paeonol was higher in the MC(H) than that in the MC(L) group (MC(L): $63.81 \pm 29.74$ vs MC(H): $4221.5 \pm 1579.2$ ng/mL, $p < 0.05$). Paeoniflorin $C_{max}$ in the MC(H) group was also higher than MC(L) group (MC(L): $60.5 \pm 15.3$ vs MC(H): $164.7 \pm 74.7$ ng/mL, $p < 0.05$). Repeated MC(H) treatments improved body weight loss and disease activity index. Western blots indicated that the expression of intestinal epithelial integrity-related proteins in the MC(H) group was comparable to that in the control. Inflammation did not influence paeonol and paeoniflorin PK significantly, whereas MC(H) group markedly increased systemic exposure, especially of paeonol, and demonstrated symptom relief. Appropriate dose adjustments are necessary to ensure safe and effective therapy because PK changes can lead to increased systemic exposure and affect treatment outcomes.

**Data availability statement:** All relevant data are within the manuscript and its Supporting Information files. Raw plasma concentration-time data files are available from the Zenodo database (DOI: 10.5281/zenodo.17462573).

**Funding:** This research was funded by the Korea Institute of Oriental Medicine, Ministry of Education, Science, and Technology, Korea [Grant No. KSN2212020]. The funders had no role in study design, data collection and analysis, decision to publish, or preparation of the manuscript.

**Competing interests:** The authors have declared that no competing interests exist.

## Introduction

Inflammatory bowel disease (IBD), which primarily includes Crohn's disease and ulcerative colitis, is a chronic, progressive, relapsing disorder characterized by intestinal mucosal inflammation [1]. Although these diseases are accompanied by various symptoms that substantially impair quality of life, such as abdominal pain, diarrhea, weight loss, and fatigue, the underlying pathogenesis remains incompletely understood. As a result, current treatment strategies rely on symptomatic management using anti-inflammatory drugs despite the adverse effects of long-term use [2]. Considering the rising global incidence of IBD and the limitations of current therapies, there is a growing need for alternative approaches that offer sustainable options for long-term disease control [3].

Moutan Cortex (MC), derived from the root bark of *Paeonia suffruticosa* Andr., is commonly used as a sedative, analgesic, and anti-inflammatory agent in traditional medicine [4]. MC has traditionally been prescribed as part of herbal formulas like Daehwang-Mudanpi-tang for treating acute appendicitis and intestinal inflammation [5,6]. Preclinical studies have demonstrated that MC effectively resolves immune cell activation and infiltration in the gut, thereby reducing colitis severity in a murine dextran sulfate sodium-induced colitis model [7]. The primary active constituents of MC, paeonol and paeoniflorin, confer anti-inflammatory and immunomodulatory effects with notable improvements in colitis [8,9]. However, the effects of pathological conditions, particularly intestinal inflammation, on the pharmacokinetic (PK) profiles of these compounds have not been fully elucidated. Given that inflammation alters drug absorption, distribution, metabolism, and excretion (ADME), understanding how these changes influence systemic exposure to active compounds is critical for rational dose adjustments and optimization of therapeutic outcomes [10,11].

This study investigated the PK profiles of paeonol and paeoniflorin under inflammatory conditions caused by dinitrobenzene sulfonic acid (DNBS)-induced colitis. Repeated oral administration was used to assess the effect of dosing frequency on the PK profiles of the main active constituents of MC and their therapeutic efficacy.

## Materials and methods

### Chemicals and reagents

A 70% ethanol extract of the MC (batch no. JE-K-6; produced in December 2022) was prepared and authenticated by the Korean Institute of Oriental Medicine (Daejeon, Republic of Korea). Reference standards, including paeonol (≥ 98.0%) and paeoniflorin (≥ 98.0%), were obtained from Sigma-Aldrich (St. Louis, MO, USA). Methanol (Sigma-Aldrich), ethyl acetate (99.5%, Samchun, Pyungtaek, Republic of Korea), and chloroform (99.5%, Samchun) were used for the extraction.

### Animals

Ten-week-old female Sprague-Dawley rats were obtained from Orient Bio (Seongnam, Republic of Korea) and acclimatized for one week before the experiments. Animals were housed under controlled environmental conditions (22–24 °C,

50–60% relative humidity) with a 12 h light/dark cycle. Animals were provided standard laboratory chow and tap water ad libitum with environmental enrichment, including nesting materials.

During the experiment, animals were monitored at least once daily for health and behavioral signs such as posture, grooming, and activity. The PK study comparing normal control (NC) and DNBS-induced groups lasted for 5 days, while the PK study comparing low- and high-dose treatment groups, as well as the efficacy evaluation involving NC, DNBS, low-, and high-dose groups, lasted for 10 days including MC administration prior to DNBS induction. Humane endpoints were including >20% body weight loss, severe lethargy, respiratory distress, or inability to access food or water. Animal that met these criteria would have been euthanized immediately using $CO_2$ chamber upon detection during scheduled monitoring (once daily); however, no animals died before reaching the predefined endpoint criteria. All personnel involved in animal experimental procedures were trained, and all procedures were approved by the Animal Care and Use Committee of the Chungnam National University (Daejeon, Republic of Korea; 202409A-CNU-179), in adherence with the Guide for the Care and Use of Laboratory Animals published by the U.S. National Institutes of Health (8th edition, revised in 2011).

## Pharmacokinetics

**Experimental design and treatment protocol.** DNBS-induced colitis model was established as previously described [12]. Briefly, 30 mg of DNBS dissolved in 250 µL of 50% ethanol was administered intrarectally using a polyurethane catheter inserted 8 cm from the anus, under isoflurane anesthesia to minimize suffering and distress. To minimize leakage, the animals were maintained in a head-down position for 3 min after injection. Control animals received the vehicle alone. PK analysis was performed on day 4 after DNBS administration, a time point selected to ensure the persistence of colonic damage, while also ensuring that the animals were sufficiently stable to undergo repeated blood sampling without severe dehydration [12,13].

The PK study consisted of two experiments. In a previous PK study, MC was administered orally at 6 g/kg for seven consecutive days in rats without adverse effects [14], confirming its safety. To confirm detectability of the major constituents, paeonol and paeoniflorin, a pilot experiment was conducted with a single 2.5 g/kg oral dose. Based on this result, in the first experiment a single oral dose of MC (2.5 g/kg in distilled water) was administered to both NC and DNBS-induced colitis groups to assess PK differences (n = 5 per group). In the second experiment DNBS-induced rats received MC orally once daily for six consecutive days at either 0.5 g/kg [MC(L)] or 2.5 g/kg [MC(H)] to evaluate the dose-dependent PK differences under inflammatory conditions (n = 5 per group). Blood samples were collected from the jugular vein at various time points (0, 0.083, 0.25, 0.5, 0.75, 1, 2, 4, 6, 8, 10, 12, and 24 h). After final sampling, the animals were euthanized. Plasma was separated by centrifugation at 12,000 rpm for 15 min and stored at −80 °C until analysis.

**Sample preparation.** Plasma MC concentrations were analyzed using ultra-performance liquid chromatography–electrospray ionization–mass spectrometry (UPLC–MS/MS, Agilent Technologies; Santa Clara, CA, USA). For paeonol extraction, 50 µL of plasma was vortexed with 500 µL of chloroform for 90 s, followed by centrifugation at 4500 rpm for 5 min at 23–25 °C. A 400-µL aliquot of the organic phase was transferred to a clean tube and evaporated to dryness under stream of nitrogen at 30 °C. For paeoniflorin extraction, 50 µL of plasma was vortexed with 500 µL of ethyl acetate and centrifuged under the same conditions. The organic layer was collected and evaporated as described above. The dried residues were reconstituted in 50 µL of methanol, vortexed, and centrifuged. A 3 µL aliquot of the resulting supernatant was injected into the UPLC–MS/MS system for analysis.

**UPLC–MS/MS analysis and method validation.** Quantitative analysis of MC constituents was performed using UPLC–MS/MS. Chromatographic separation was achieved using a Hypersil GOLD™ C18 column (2.1×100 mm, 3 µm; Thermo Scientific, Waltham, MA, USA) maintained at 40 °C. The mobile phase consisted of 0.1% formic acid in distilled water (A) and 0.1% formic acid in acetonitrile (B) delivered at a rate of 0.3 mL/min under the following gradient flow: 0–5 min, 85–30% A; 5–7 min, 30–85% A; and 7–9 min, 85% A.

 

Paeonol was detected in positive ionization mode at $m/z$ 167→121 with a collision energy of 20 eV, whereas paeoniflorin was detected in negative ionization mode at $m/z$ 525→449 with a collision energy of 10 eV. The MS/MS parameters for both analytes were: gas temperature, 275 °C; gas flow, 5 L/min; nebulizer pressure, 50 psi; sheath gas temperature, 250 °C; and sheath gas flow, 11 L/min.

The linearity of the method ($r^2$) was validated using blank rat plasma samples spiked with six concentration levels (0, 0.01, 0.05, 0.1, 0.5, 1 µg/mL) for both paeonol and paeoniflorin, based on their peak area ratios. Intra-day accuracy and precision were assessed in triplicate within a 24 h period, and inter-day precision was evaluated across three consecutive days. The limit of quantification (LOQ) was defined as the lowest concentration yielding acceptable accuracy (80–120%) and precision (<15%). The limits of detection (LOD) and quantification (LOQ) for plasma concentration analysis were determined based on analyte concentrations corresponding to signal-to-noise (S/N) ratios of 3 and 10, respectively, following the "Guideline on Bioanalytical Method Validation" from the Ministry of Food and Drug Safety in Korea (MFDS, 2023). The detailed validation results for both paeonol and paeoniflorin are presented in S1 Table.

**Pharmacokinetic and statistical analysis.** The raw concentration-time data for all individual animals are available from the Zenodo database (https://doi.org/10.5281/zenodo.17462573). The PK parameters were calculated using a non-compartmental model using Monolix 2023R1 (Lixoft SAS, Simulations Plus Inc., Lancaster, CA, USA). The dose administered for each component was determined based on its MC content and was applied accordingly in the PK analysis. The maximum plasma concentration ($C_{max}$) and time to reach $C_{max}$ ($T_{max}$) were obtained directly from the observed data. The elimination half-life was estimated using linear regression of the log-transformed concentration-time data in the terminal phase. The area under the curve (AUC) was calculated using the linear up/log-down trapezoidal method to determine the final concentration of each group.

## Efficacy evaluation

**Experimental design and treatment protocol.** Ten-week-old female Sprague-Dawley rats (Orient Bio, Sungnam, Republic of Korea) were randomly assigned to four groups (n = 5 per group): NC, DNBS (DNBS induction without treatment), MC(L) (low-dose MC, 0.5 g/kg/day for 6 days + DNBS), and MC(H) (high-dose MC, 2.5 g/kg/day for 6 days + DNBS).

**Body weight and evaluation of colon damage.** The animals were weighed daily from day 0 until sacrifice. The disease activity index (DAI) score (0–4) was assessed daily based on three parameters: body weight loss relative to baseline, stool consistency, and presence of bleeding [15]. On day 7 following euthanasia, the colon was collected, and the length (cm) of each specimen was measured. All tissue samples were stored at −80 °C for subsequent biochemical analyses.

**Histological analysis.** Colon tissues were fixed in 10% neutral-buffered formalin, embedded in paraffin, and sectioned at 4 µm thickness using a HistoCore BIOCUT microtome (149BIO000C1, Leica Biosystems, Cambridge, UK). Sections were stained with hematoxylin and eosin (H&E; TissuePro Technology, Gainesville, FL, USA) and periodic acid-Schiff (PAS; Sigma-Aldrich), according to the manufacturer's protocols. For each slide, ten non-overlapping fields were randomly selected and imaged using a slide scanner (Motic, Schertz, TX, USA).

**Western blot analysis.** Total proteins were extracted by homogenizing 10 mg of rat colon tissue in 1 mL of CelLytic™ MT Cell Lysis Reagent (Sigma) supplemented with phosphatase (Roche, Basel, Switzerland) and protease inhibitors (Roche), using a BIOPREP-24R homogenizer (Allsheng Instruments Co., Hangzhou, China). Protein concentrations were determined using a bicinchoninic acid assay (Thermo Fisher Scientific). Equal amounts of protein (20 µg per sample) were separated on 6–10% sodium dodecyl sulphate–polyacrylamide gel electrophoresis gels and transferred to polyvinylidene difluoride membranes (Millipore Sigma, Burlington, MA, USA).

Membranes were blocked with 5% bovine serum albumin in tris-buffered saline with Tween 20 for 1 h at 23–25 °C, followed by overnight incubation at 4 °C with the following primary antibodies (all at 1:1000 dilution): TNF-α (abcam, Cambridge, UK), COX-2 (abcam), TGF-β1 (abcam), AMPK (Cell signaling, Danvers, MA, USA), Occludin (abcam), MMP-9

(Novus Biologicals, Centennial, CO, USA), and β-actin (Cell signaling). After washing the membranes three times for 10 min each with phosphate buffered saline with Tween 20, they were incubated for 2 h at 23–25 °C with horseradish peroxidase-conjugated secondary antibodies (1:5000; Abfrontier Co., Ltd., Seoul, Republic of Korea). After three additional washes, protein bands were visualized using enhanced chemiluminescence and imaged using a ChemiDoc imaging system (Bio-Rad Laboratories, Hercules, CA, USA). Band intensities were quantified using ImageJ software, version 1.49 (NIH, Bethesda, MD, USA) and normalized to β-actin. Full-length, uncropped Western blot images for all targets are provided in the S1 Fig. with molecular weight markers.

**Statistical analysis.** All data are presented as mean ± standard deviation (SD). The Shapiro–Wilk test was used to assess data normality. One-way analysis of variance was performed for normally distributed data with homogeneity of variance, followed by Bartlett's test to verify equal variance. Post-hoc comparisons between groups were conducted accordingly. In cases of non-normal distribution or unequal variances, the non-parametric Kruskal–Wallis H test was applied. Statistical significance was set at $*p < 0.05$, $**p < 0.01$, and $***p < 0.001$. All statistical analyses were performed using GraphPad Prism, version 8 (GraphPad Software, Boston, MA, USA).

## Results

### Analysis of active constituents in MC

By comparing the retention times and peak matching with those of the reference standards, paeonol and paeoniflorin were identified in MC. Quantification was performed using an external standard calibration method. Paeonol and paeoniflorin accounted for approximately 6% and 1.8% of the total MC weight, respectively. Fig 1A shows the UPLC–MS/MS analysis of the 70% hydro-ethanolic extract of MC, and Fig 1B shows the chromatograms of the corresponding reference substances.

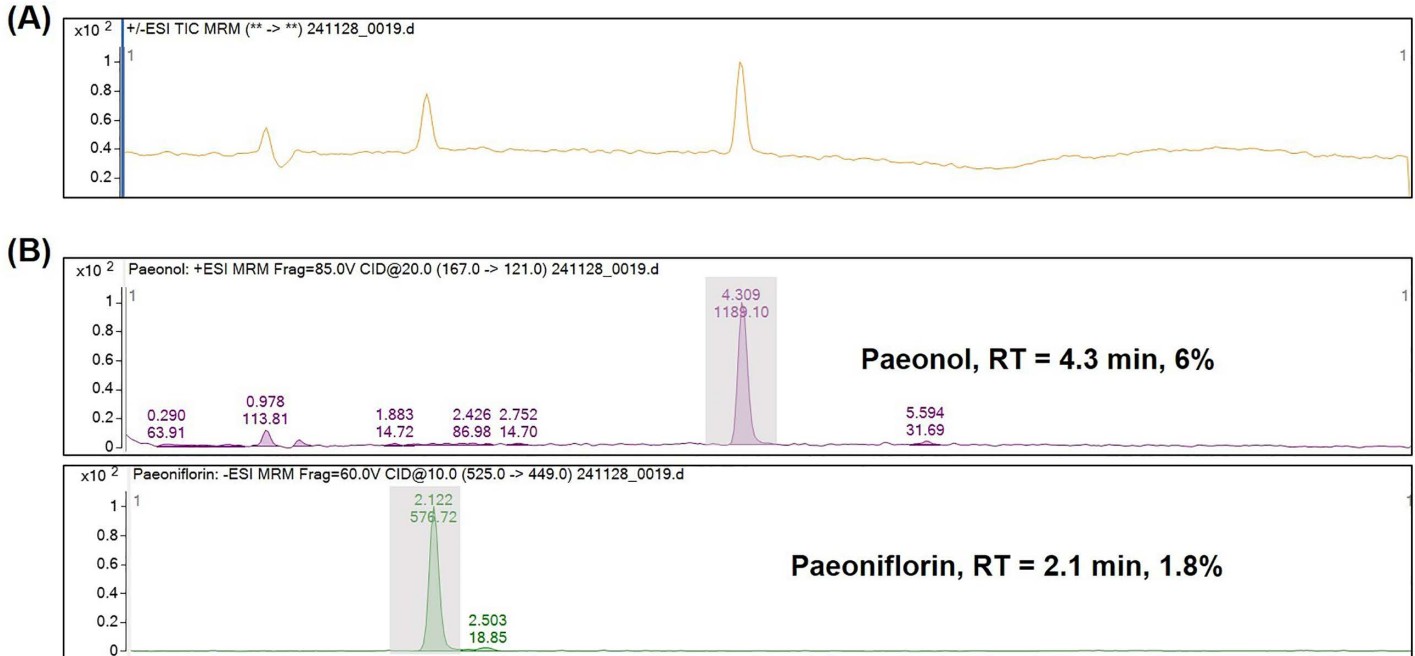

**Fig 1. Ultra-performance liquid chromatography–electrospray ionization–mass spectrometry analysis of Moutan Cortex extract. (A)** Total ion chromatogram of the 70% hydroethanolic extract of MC. **(B)** Multiple reaction monitoring chromatograms of reference standards. The relative content of paeonol (RT = 4.3 min) and paeoniflorin (RT = 2.1 min) in the MC extract was estimated to be 6.0% and 1.8%, respectively, of the total chromatographic peak area.

## Pharmacokinetics of MC in NC and DNBS-treated group

The plasma concentration-time curves illustrate the PK profiles of paeoniflorin and paeonol in the NC and DNBS-treated groups (Fig 2). There were no statistically significant differences in the plasma concentrations of either compound between the two groups. The PK parameters are listed in Table 1, and individual concentration-time profiles are provided in S2 Fig.

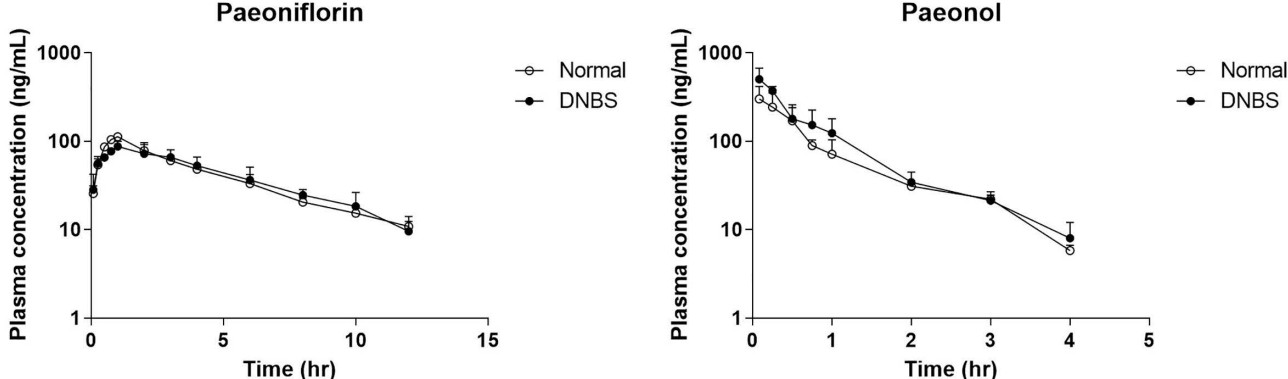

**Fig 2. Mean plasma concentration-time profiles of paeoniflorin and paeonol after single oral administration in NC and DNBS-induced colitis rats.** The NC group received a single oral dose of MC at 2.5 g/kg without prior treatment. The DNBS group was pretreated with DNBS to induce colitis and received the same MC dosage. Values are expressed as mean ± SD (n = 5).

**Table 1. PK profiles of paeoniflorin and paeonol after single oral administration in NC and DNBS-induced colitis rats.**

| Compound | Parameter | Unit | NC | DNBS |
|---|---|---|---|---|
| Paeoniflorin | Rsq | | 0.97 ± 0.02 | 0.92 ± 0.03 |
| | HL_Lambda_z | hr | 4.2 ± 1.3 | 3.0 ± 1.1 |
| | $T_{max}$ | hr | 1 | 1 (0.75-1) |
| | $C_{max}$ | ng/mL | 112.4 ± 5.5 | 87.3 ± 12.5 |
| | $AUC_{last}$ | hr*ng/mL | 474.1 ± 11.7 | 463.7 ± 106.8 |
| | AUC_%Extrap_obs | % | 12.4 ± 6.6 | 7.4 ± 6.6 |
| | $AUMC_{last}$ | hr*hr*ng/mL | 1791.9 ± 119.8 | 1814.4 ± 532.3 |
| | $MRT_{last}$ | hr | 3.7 ± 0.2 | 3.9 ± 0.5 |
| Paeonol | Rsq | | 0.93 ± 0.01 | 0.98 ± 0.01 |
| | HL_Lambda_z | hr | 0.9 ± 0.1 | 0.7 ± 0.2 |
| | $T_{max}$ | hr | 0.08 | 0.08 (0.08-0.25) |
| | $C_{max}$ | ng/mL | 300.7 ± 114.8 | 503.1 ± 166.9 |
| | $AUC_{last}$ | hr*ng/mL | 247.8 ± 63.7 | 337.0 ± 120.8 |
| | AUC_%Extrap_obs | % | 5.7 ± 3.9 | 3.7 ± 2.4 |
| | $AUMC_{last}$ | hr*hr*ng/mL | 223.4 ± 5.33 | 270.0 ± 128.5 |
| | $MRT_{last}$ | hr | 0.9 ± 0.2 | 0.7 ± 0.1 |

The data represented means ± SD (n = 5). AUC_%Extrap_obs, area under the curve percentage extrapolated (Observed); $AUC_{last}$, area under the curve after the last dose; $AUMC_{last}$, area under the first moment curve from time zero to the last measurable concentration; $C_{max}$, maximum plasma concentration of a drug; DNBS, dinitrobenzene sulfonic acid; HL_Lambda_z, half-life Lambda z; MC, Moutan Cortex; $MRT_{last}$, mean residence time to the last measurable concentration; PK, pharmacokinetic; Rsq, R-squared; $T_{max}$, time to reach $C_{max}$.

For paeoniflorin, the $C_{max}$ was slightly lower in the DNBS groups (87.3 ± 12.5 ng/mL) than in the NC (112.4 ± 5.5 ng/mL), with $AUC_{last}$ values of 463.7 ± 106.8 and 474.1 ± 11.7 hr*ng/mL, respectively. Other parameters, including half-life (HL_Lambda_z) and mean residence time ($MRT_{last}$), remained relatively unchanged between groups.

In the case of paeonol, the DNBS group exhibited a higher $C_{max}$ (503.1 ± 166.9 ng/mL) than the NC (300.7 ± 114.8 ng/mL), and $AUC_{last}$ also appeared elevated (337.0 ± 120.8 vs. 247.8 ± 63.7 hr*ng/mL). However, this difference was not statistically significant. When comparing the two compounds within the NC group, paeonol showed a higher and earlier peak, with a $C_{max}$ of 300.7 ± 114.8 ng/mL at 0.08 h, whereas paeoniflorin peaked later at 1 h with a lower $C_{max}$ of 112.4 ± 5.5 ng/mL. However, paeonol exhibited a shorter half-life (0.9 vs. 4.2 h) and lower systemic exposure ($AUC_{last}$: 247.8 vs. 474.1 hr*ng), indicating a more rapid elimination profile than paeoniflorin.

### Evaluation of PK analysis after repeated administration of MC

The PK parameters of MC were evaluated at two doses (0.5 and 2.5 g/kg) following repeated oral administration for six consecutive days in DNBS-induced colitis in rats (Fig 3). The PK profiles are summarized in Table 2, and individual concentration-time profiles are provided in S3 Fig.

For paeoniflorin, the MC(H) group demonstrated higher $C_{max}$ (164.7 ± 74.7 ng/mL) and $AUC_{last}$ (867.6 ± 383.8 hr*ng/mL) than the MC(L) group, which showed a $C_{max}$ of 60.5 ± 15.3 ng/mL and $AUC_{last}$ of 201.0 ± 71.4 hr*ng/mL) (p < 0.05). The $MRT_{last}$ was longer in the MC(H) (4.3 ± 0.1 h) than the MC(L) group (2.5 ± 0.8 h, p < 0.05), indicating a dose-dependent increase in systemic exposure.

For paeonol, the differences between groups were even more pronounced. The MC(H) group showed a markedly elevated $C_{max}$ (4221.5 ± 1579.2 ng/mL), which was more than 60 times higher than that of the MC(L) group (63.81 ± 29.74 ng/mL, p < 0.05). In the MC(L) group, the PK assessment of paeonol was restricted because of insufficient quantifiable plasma concentrations at subsequent time points. In contrast, the MC(H) group exhibited rapid absorption ($T_{max}$, 0.25 h) and a relatively short elimination half-life (HL_Lambda_z: 1.3 ± 0.0 h). These findings indicate distinct dose-dependent changes in the PK of both compounds, with more significant differences observed for paeonol than for paeoniflorin.

### Effects of MC treatment on symptoms in DNBS-induced colitis model

All groups administered DNBS exhibited body weight loss starting on day 1 post-administration (Fig 4A). By day 5, all DNBS-treated groups experienced slight weight gain. The DAI scores increased from day 1 in all DNBS-treated groups,

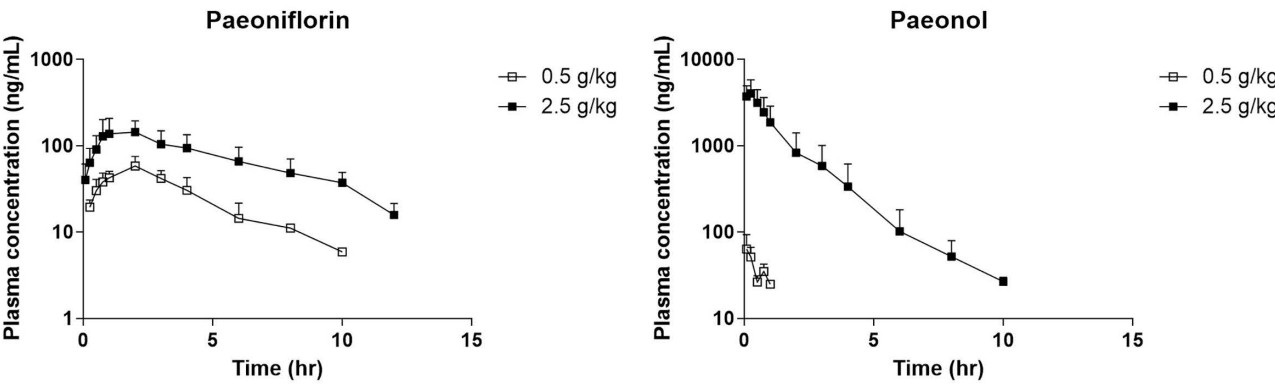

**Fig 3. Mean plasma concentration-time profiles of paeoniflorin and paeonol comparing low and high repeated doses of MC in DNBS-induced colitis rats.** The MC(L) group received 0.5 g/kg of MC daily for six consecutive days, whereas the MC(H) group received 2.5 g/kg under the same schedule. Values are expressed as mean ± standard deviation (SD) (n = 5).

**Table 2. PK profiles of paeoniflorin and paeonol comparing low and high repeated doses of MC in DNBS-induced colitis rats.**

| Compound | Parameter | Unit | MC(L) (0.5 g/kg) | MC(H) (2.5 g/kg) |
|---|---|---|---|---|
| Paeoniflorin | Rsq | | $0.98 \pm 0.03$ | $0.92 \pm 0.07$ |
| | HL_Lambda_z | hr | $1.4 \pm 0.6^*$ | $4.1 \pm 2.1$ |
| | $T_{max}$ | hr | 2 (1-2) | 1.5 (1-2) |
| | $C_{max}$ | ng/mL | $60.5 \pm 15.3^*$ | $164.7 \pm 74.7$ |
| | $AUC_{last}$ | hr*ng/mL | $201.0 \pm 71.4^*$ | $867.6 \pm 383.8$ |
| | AUC_%Extrap_obs | % | $9.9 \pm 4.16$ | $9.8 \pm 7.7$ |
| | $AUMC_{last}$ | hr*hr*ng/mL | $555.0 \pm 386.2^*$ | $3789.4 \pm 1650.7$ |
| | $MRT_{last}$ | hr | $2.5 \pm 0.8^*$ | $4.3 \pm 0.1$ |
| Paeonol | Rsq | | – | $0.96 \pm 0.04$ |
| | HL_Lambda_z | hr | – | $1.3 \pm 0.0$ |
| | $T_{max}$ | hr | 0.08 (0.08-0.5) | 0.25 (0.08-0.25) |
| | $C_{max}$ | ng/mL | $63.81 \pm 29.74^*$ | $4221.5 \pm 1579.2$ |
| | $AUC_{last}$ | hr*ng/mL | – | $5780.9 \pm 3656.7$ |
| | AUC_%Extrap_obs | % | – | $1.1 \pm 0.6$ |
| | $AUMC_{last}$ | hr*hr*ng/mL | – | $9191.6 \pm 7408.1$ |
| | $MRT_{last}$ | hr | – | $1.4 \pm 0.3$ |

Significance: *$p < 0.05$, compared to the high dose group. The data represented means $\pm$ SD (n = 5). AUC_%Extrap_obs, area under the curve percentage extrapolated (Observed); $AUC_{last}$, area under the curve after the last dose; $AUMC_{last}$, area under the first moment curve from time zero to the last measurable concentration; $C_{max}$, maximum plasma concentration of a drug; DNBS, dinitrobenzene sulfonic acid; HL_Lambda_z, half-life Lambda z; MC, Moutan Cortex; $MRT_{last}$, mean residence time to the last measurable concentration; PK, pharmacokinetic; Rsq, R-squared; $T_{max}$, time to reach $C_{max}$

indicating symptom aggravation, but gradually decreased from day 4 onward. The MC(H) group showed a marked reduction in the DAI by day 5 (Fig 4B).

Macroscopic observations indicated a substantial decrease in colon length in the DNBS group, whereas the colon length in the MC-treated groups was restored to levels comparable to those in the NC (Fig 4C). Quantitative analysis confirmed a shortening of the colon of approximately 27% in the DNBS group compared to the NC ($p < 0.01$). In comparison, the MC(H) group exhibited an increase in colon length of approximately 23% compared to the DNBS group ($p < 0.05$) (Fig 4D).

H&E staining demonstrated epithelial disruption and infiltration of inflammatory cells in the DNBS group, which were alleviated in all MC-treated groups (Fig 4E). Periodic acid–Schiff staining indicated a marked reduction in the number of mucus-secreting cells in the DNBS group. This reduction was ameliorated in the MC(H) group, which showed a more pronounced recovery of mucus secretion than that in the MC(L) group.

### Effects of MC treatment on intestinal epithelial improvement in DNBS-induced colitis model

Intestinal epithelial damage is a hallmark of IBD and is characterized by impaired barrier function and an enhanced inflammatory response due to the disruption of tight junctions [16]. In the present study, levels of early inflammatory markers, such as TNF-α and COX-2, were examined initially. There were no significant differences between the NC and DNBS-treated groups (Fig 5A), suggesting that the model progressed beyond the acute inflammatory phase. Therefore, we focused on the factors involved in tissue repair mechanisms. To investigate mucosal repair, the expression of key regulatory factors, including TGF-β1, AMPK, and occludin, was examined in the DNBS-induced IBD model. AMPK expression was elevated in the DNBS group compared to the NC group (1.8-fold, $p < 0.001$), but was markedly reduced in both the MC(L) and MC(H) groups relative to the DNBS group ($p < 0.001$ for both) (Fig 5B). TGF-β1 expression in the DNBS group was approximately 1.5-fold higher than in NC (ns), whereas MC(L) and MC(H) groups exhibited 40% ($p < 0.05$) and 50% ($p < 0.01$) reductions compared to DNBS, respectively.

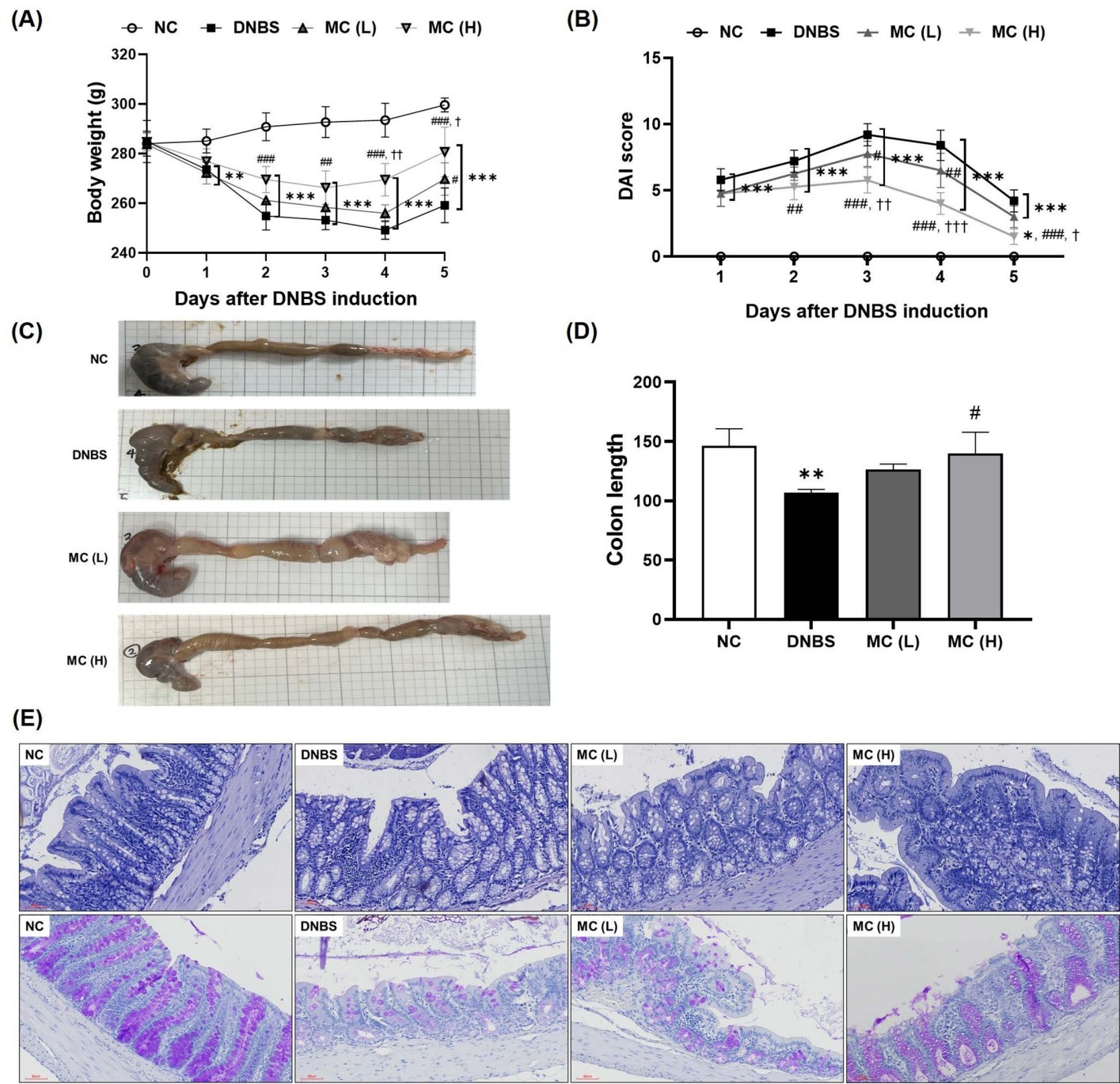

**Fig 4. Effects of MC treatment on symptoms and histological changes in DNBS-induced colitis rats.** Effects of MC treatment on symptoms and histological changes in rats with DNBS-induced colitis. **(A)** Body weight changes (g) over time after DNBS induction. **(B)** Disease activity index (DAI) score over time after DNBS induction, **(C)** Representative macroscopic images of colons from each group. **(D)** Colon length measurements. **(E)** Representative histological images: upper panels show hematoxylin and eosin staining; lower panels show periodic acid–Schiff staining (scale bar = 60 μm for all images). The data represented means ± SD (n = 5). Statistical significance: *p < 0.05, **p < 0.01, and ***p < 0.001 vs. NC; #p < 0.05, ##p < 0.01, and ###p < 0.001 vs. DNBS; †p < 0.05, ††p < 0.01, and †††p < 0.01 vs. MC(L).

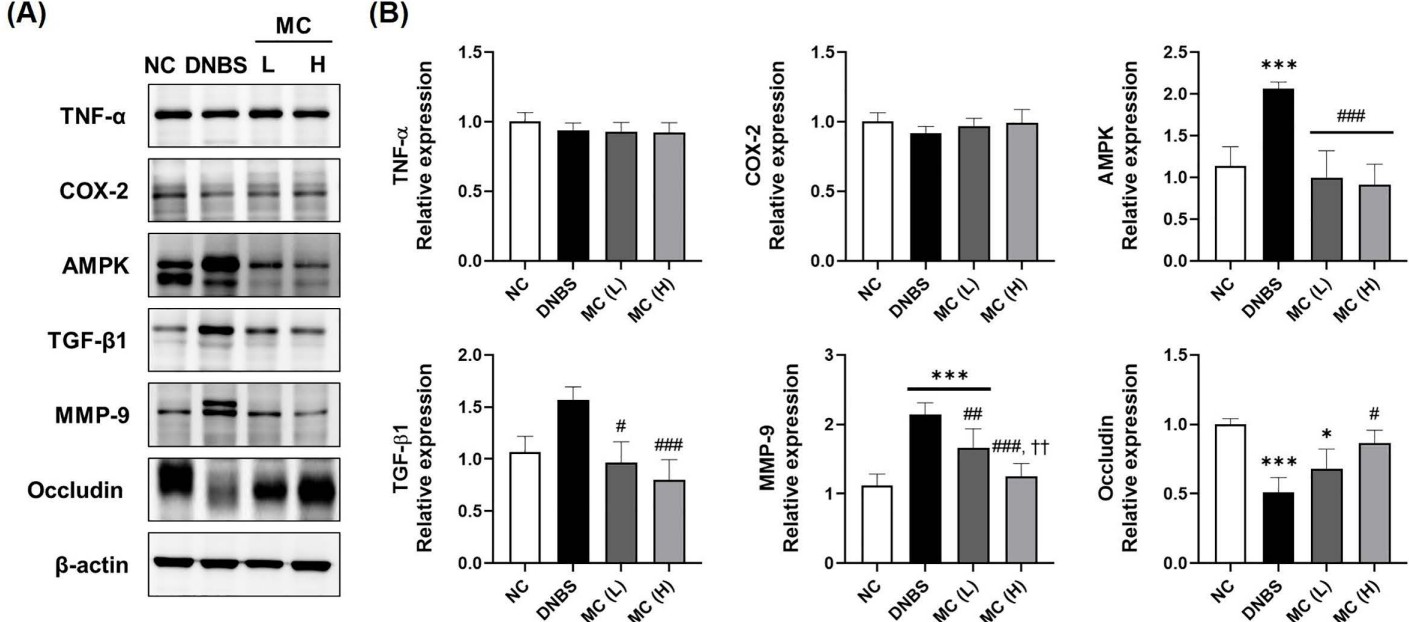

**Fig 5. Effects of MC treatment on protein expression in DNBS-induced colitis rats. (A)** Western blot analysis of colon tissue for the expression of TNF-α, COX-2, AMPK, TGF-β1, MMP-9, and occludin. **(B)** Densitometric quantification of the relative expression of each protein, normalized to β-actin. The data represented means ± standard deviation (SD) (n = 5). *p < 0.05 and ***p < 0.001 vs NC group; #p < 0.05, ##p < 0.01, and ###p < 0.001 vs DNBS group; ††p < 0.01 vs. MC(L).

MMP-9, which degrades tight junction proteins and compromises the mucosal barrier integrity [17], was also examined. The DNBS group showed a two-fold increase in MMP-9 expression compared to NC (p < 0.001), whereas the MC(L) and MC(H) groups showed approximately 30% (p < 0.01) and 40% (p < 0.001) reductions relative to the DNBS group, respectively. Moreover, the MMP-9 expression in the MC(H) group was lower than that in the MC(L) group (approximately 25% reduction, p < 0.01). Additionally, occludin expression was reduced by approximately 50% in the DNBS group compared to NC (p < 0.001). The MC(H) group demonstrated a 1.7-fold increase relative to the DNBS group (p < 0.05), indicating enhanced restoration of tight junction integrity.

## Discussion

In this study, we investigated the effects of MC administration in a colitis model. MC has been used to treat inflammation in China and other Southeast Asian countries [4]. We evaluated the PK properties of its constituents, disease-induced alterations in absorption, the impact of repeated dosing, and observed therapeutic efficacy.

We confirmed that colitis had minimal impact on the pharmacokinetic profiles of both paeonol and paeoniflorin. This finding suggests that the inflammatory condition itself does not substantially alter systemic exposure of MC. Our results align with previous reports showing that the pharmacokinetics of several anti-inflammatory agents do not significantly differ between healthy individuals and patients with IBD [18]. Therefore, any observed changes in systemic exposure are unlikely to result from the inflammatory state but rather from dosing conditions.

To investigate this, we next evaluated the effects of repeated oral administration under colitis conditions. Considering the active constituents in MC, the administration of 2.5 g/kg corresponded to approximately 150 mg/kg paeonol and 45 mg/kg paeoniflorin. Given their respective contents and oral bioavailability values (~15% for paeonol and ~4% for paeoniflorin) [19], paeonol was expected to exhibit higher plasma concentrations and greater systemic exposure.

As expected, paeonol exhibited a higher average $C_{max}$ and a shorter $T_{max}$ than paeoniflorin. However, the elimination half-life and MRT were shorter, with a lower AUC. These findings highlight the limited systemic exposure to paeonol, despite its higher dose and greater oral bioavailability, reflecting differences in clearance and intrinsic structural and metabolic properties of the two compounds. Moreover, in the low-dose (0.5 g/kg) repeated-dose group, paeonol plasma concentrations were insufficient for PK analysis, whereas in the high-dose (2.5 g/kg) repeated-dose group, systemic exposure was markedly increased. Both $C_{max}$ and AUC exceeded those observed after single-dose administration by more than 10-fold.

Paeoniflorin and paeonol exhibit distinctly different chemical and metabolic characteristics, which underscores their markedly different absorption and systemic behaviors. Paeoniflorin is a hydrophilic monoterpene glycoside (MW: 480.46 g/mol, LogP: −1.2) with low lipophilicity and large molecular size, resulting in limited paracellular absorption [20,21]. Additionally, it is subject to active efflux by transporters such as P-glycoprotein, which further reduces its absorption rate [22,23]. Paeoniflorin undergoes extensive metabolism by the intestinal microbiota, with a relatively minor contribution from hepatic metabolism [24,25]. For these reasons, paeoniflorin has very low oral bioavailability, limiting its systemic exposure.

Paeonol is a small, lipophilic molecule (MW: 166.17 g/mol, LogP: 1.98) that is primarily absorbed via transcellular pathways [19,26]. In the present study, paeonol exhibited a $T_{max}$ of 0.08 h, consistent with previous reports of rapid absorption (typically 0.1–0.2 h). Following absorption, paeonol undergoes extensive first-pass metabolism in the liver primarily through glucuronidation and sulfation, leading to rapid systemic clearance and a short elimination half-life [27]. Repeated high-dose administration resulted in increased systemic exposure compared to single dosing, suggesting that the oral bioavailability of paeonol may increase under such conditions, possibly due to the partial saturation of metabolic clearance pathways. In contrast, the oral bioavailability of paeoniflorin remained consistently low, reflecting its limited permeability and strong dependence on efflux transporters, regardless of dosing frequency. Therefore, the pharmacokinetic changes observed in this study were primarily attributable to factors such as repeated administration or high dosage. Although paeonol has been recognized as a relatively safe substance [19], caution is advised when it is administered repeatedly at high doses as it may lead to increased systemic exposure beyond that observed with a single dose.

Given these distinct PK profiles, we evaluated whether differences in systemic exposure were associated with therapeutic efficacy in a DNBS-induced colitis model. In this study, the DAI was highest on day 3 after DNBS induction, followed by a gradual decline by day 5. The observed weight gain, DAI reduction, and increased colon length in the MC group suggest the therapeutic effectiveness of MC. Histopathological evaluation indicated that MC preserved the intestinal epithelial integrity and inhibited the reduction of goblet cells. MC reduced the expression of proteins (MMP-9, TGF-β1, AMPK) that were elevated due to DNBS. TGF-β regulates the expression of tight junction proteins and adhesion molecules, which play a crucial role in modulating and enhancing epithelial barrier function [28]. AMPK also plays a vital role in epithelial repair [29]. MMP-9 compromises tight junctions between mucosal cells, leading to increased intestinal mucosal permeability and weakened mucosal barrier function [17]. Therefore, the decrease of these proteins due to MC implies that the damage caused by DNBS was mitigated in the MC group, indicating a protective role through the reduced expression of TGF-β1 and AMPK compensatory pathways, which could improve epithelial integrity and facilitate repair. The concentration-dependent restoration of occludin expression to NC levels in the MC group further suggests that MC supports the repair of tight junctions between intestinal epithelial cells and enhances barrier function.

Collectively, these findings support the protective potential of MC against DNBS-induced intestinal injury. However, the overall interpretation of these effects should be made with caution due to the small group size (n = 5) used in this study. Among the measured pharmacokinetic parameters, only a few variables, such as $C_{max}$, showed statistically significant differences, suggesting that the effect size for these endpoints was large enough to be detected with five animals per group. For other parameters showing greater interindividual variability, however, the small sample size likely limited the statistical power of the analysis.

In addition, although systemic paeonol exposure markedly increased after six consecutive days of high-dose treatment, no clinical or behavioral signs of toxicity were observed. Nevertheless, biochemical assessments of hepatic or renal function and histopathological examinations of organs other than the colon were not performed, which should be recognized as a limitation of this study.

## Conclusions

This comprehensive evaluation demonstrated that colitis did not markedly affect the disposition of paeonol and paeoniflorin, indicating that symptomatic improvement was primarily influenced by dosing regimens and repeated administration. Therefore, the repeated-dosing pharmacokinetic assessment compared low and high-dose administration only in the DNBS-induced colitis model, without including a normal control group. These findings, together with the results of the present study, support the notion that dosing regimens can be considered in inflammatory states when systemic pharmacokinetics are not markedly affected. Nevertheless, further studies with extended observation periods and the inclusion of normal controls are needed to confirm these results and more clearly delineate potential variability.

## Supporting information

**S1 Fig. Original full-length gel image related to Fig 5. Full, non-adjusted gel image corresponding to the Western blot data shown in Fig 5. Molecular weight markers are indicated in kilodaltons (kDa).** AMPK, AMP-activated protein kinase; COX-2, cyclooxygenase-2; MMP-9, matrix metallopeptidase-9; TGF-β1, transforming growth factor-beta 1; TNF-α, tumor necrosis factor-alpha.
(JPG)

**S2 Fig. Individual concentration-time profiles of paeoniflorin and paeonol after single oral administration in NC and DNBS-induced colitis rats.** Individual plasma concentration-time curves for paeoniflorin and paeonol following a single oral administration of MC (2.5 g/kg) in NC and DNBS-induced colitis rats (n = 5). DNBS, dinitrobenzene sulfonic acid; MC, Moutan Cortex; NC, normal control.
(JPG)

**S3 Fig. Individual concentration-time profiles of paeoniflorin and paeonol comparing low and high repeated doses of MC in DNBS-induced colitis rats.** Individual plasma concentration-time profiles for paeoniflorin and paeonol after repeated oral dosing for six consecutive days with low (0.5 g/kg) or high (2.5 g/kg) MC in DNBS-treated rats (n = 5). DNBS, dinitrobenzene sulfonic acid; MC, Moutan Cortex.
(JPG)

**S1 Table. Validation results for the analysis method of paeonol and paeoniflorin.** CV, coefficient of variation; LOD, limit of detection; LOQ, limit of quantification; QC, quality control.
(DOCX)

## Acknowledgments

We would like to appreciate the Korean Institute of Oriental Medicine (Daejeon, Republic of Korea) for the preparation and authentication of the 70% ethanol extract of MC (batch no. JE-K-6).

## Author contributions

**Conceptualization:** Je-Won Ko, Tae-Won Kim.

**Data curation:** Jin-Hwa Kim, Jeong-Won Kim, Eun-Hye Chung, Su-Ha Lee.

**Formal analysis:** Jin-Hwa Kim, Ji-Soo Jeong, Jeong-Won Kim.

**Funding acquisition:** Youn-Hwan Hwang.

**Investigation:** Jin-Hwa Kim, Ji-Soo Jeong, Jeong-Won Kim, Eun-Hye Chung, Su-Ha Lee.

**Methodology:** Jin-Hwa Kim, Ji-Soo Jeong, Jeong-Won Kim, Eun-Hye Chung, Su-Ha Lee.

**Project administration:** Je-Won Ko, Tae-Won Kim.

**Resources:** Youn-Hwan Hwang, Tae-Won Kim.

**Software:** Jin-Hwa Kim.

**Supervision:** Je-Won Ko, Youn-Hwan Hwang, Tae-Won Kim.

**Visualization:** Jin-Hwa Kim, Ji-Soo Jeong, Eun-Hye Chung, Su-Ha Lee.

**Writing – original draft:** Jin-Hwa Kim.

**Writing – review & editing:** Je-Won Ko, Youn-Hwan Hwang, Tae-Won Kim.

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
