## [Decision Letter · Decision Letter 0]

30 Sep 2025

Dear Dr. Kim,

Thank you for submitting your manuscript to PLOS ONE. After careful consideration, we feel that it has merit but does not fully meet PLOS ONE’s publication criteria as it currently stands. Therefore, we invite you to submit a revised version of the manuscript that addresses the points raised during the review process.

Note from the Academic Editor: Please note that I have acted as a reviewer for this manuscript, and you will find my comments below, under Reviewer 1.

We look forward to receiving your revised manuscript.

Kind regards,

Zheng Yuan

Academic Editor

PLOS ONE

Journal Requirements:

“The authors declare that there are no conflicts of interest that could have appeared to influence the work reported in this paper.”

“The authors declare that there are no conflicts of interest that could have appeared to influence the work reported in this paper.”

Reviewers' comments:

Reviewer's Responses to Questions

**Comments to the Author**

1. Is the manuscript technically sound, and do the data support the conclusions?

Reviewer #1: Yes

Reviewer #2: Partly

2. Has the statistical analysis been performed appropriately and rigorously?

Reviewer #1: Yes

Reviewer #2: Yes

3. Have the authors made all data underlying the findings in their manuscript fully available?

Reviewer #1: Yes

Reviewer #2: No

4. Is the manuscript presented in an intelligible fashion and written in standard English?

Reviewer #1: Yes

Reviewer #2: Yes

Reviewer #1: The authors of the manuscript focused on not only profiling the PKs of two chemicals from Moutan Cortex after single and repeated administration but also detailed the pharmacology process through molecular biological methods. It deserves stress that the authors adopted repeated administration and discovered the exposure differences of the two chemicals. The manuscript will help understand pharmacology and provide a good model for profiling the PKs of multiplex constitutes of herbal medicine. What is more, the manuscript was written well, and the workload is also enough. The manuscript is recommended to be accepted by the journal PloS One. Here are several minor concerns for the authors to improve the quality of the manuscript:

1) In the Abstract section, the authors provided some results with fold. If the authors can provide number comparisons such as “number vs. number”, will help readers understand the work of the manuscript more directedly.

2) In the Keywords section, the last keyword should be deleted. And the word “pharmacokinetic” should be moved as the last keywords.

3) In line 67 and ling 68, the abbreviation ADME should be added.

4) As the authors used animal experiments, ethical issues should be provided or exhibited.

5) When introducing the western blot analysis procedure, the version of the ImageJ software should be provided.

Reviewer #2: Major concerns:

1. The manuscript inconsistently refers to “mice” in some sections (abstract/introduction) while the Methods clearly state that Sprague-Dawley rats were used. This is a fundamental error and must be corrected throughout the text for clarity and accuracy.

2. Each group contains only five animals, which is relatively small for pharmacokinetic and histological comparisons, especially given the large interindividual variability observed (e.g., paeonol Cmax at high dose). The authors should either increase the sample size, provide power calculations, or at minimum show individual concentration–time profiles to illustrate variability.

3. Please provide raw PK data (machine-readable) as mentioned. Reference them clearly in the main text.

Minor concern:

1. Given the dramatic increase in paeonol systemic exposure at the high dose, safety/toxicity monitoring is important. The manuscript currently states that no euthanasia endpoints were met, but it does not report biochemical markers (e.g., liver/kidney function) or histopathological observations beyond the colon. These should be included or at least discussed as a limitation.

**Do you want your identity to be public for this peer review?** For information about this choice, including consent withdrawal, please see our Privacy Policy

Reviewer #1: No

Reviewer #2: **Yes: ** Zixu Wang

---

## [Author Response · Author response to Decision Letter 1]

31 Oct 2025

Reviewer 1.

1) In the Abstract section, the authors provided some results with fold. If the authors can provide number comparisons such as “number vs. number”, will help readers understand the work of the manuscript more directedly.

We agree with this comment. To help readers understand the differences more intuitively, we revised the text to include the exact numerical values instead of fold changes (Line 34–35 and 37–39).

2) In the Keywords section, the last keyword should be deleted. And the word “pharmacokinetic” should be moved as the last keywords.

We appreciate the thoughtful suggestion. We have removed the last key word and repositioned “pharmacokinetic” as the final keyword.

3) In line 67 and ling 68, the abbreviation ADME should be added.

Thank you for the comment. We have added the abbreviation “ADME” to the text (Line 68-69).

4) As the authors used animal experiments, ethical issues should be provided or exhibited.

Thank you for the valuable comment. We have provided detailed information on ethical approval and considerations with the laboratory animal care guidelines (Line 101-105).

5) When introducing the western blot analysis procedure, the version of the ImageJ software should be provided.

Thank you for pointing that out. We have added the ImageJ software version used for densitometric analysis in the Methods part (Line 209).

Reviewer 2.

1) The manuscript inconsistently refers to “mice” in some sections (abstract/introduction) while the Methods clearly state that Sprague-Dawley rats were used. This is a fundamental error and must be corrected throughout the text for clarity and accuracy.

Thank you for pointing this out. We carefully reviewed the entire manuscript and corrected all to accurately refer to rats instead of mice (Line 31).

2) Each group contains only five animals, which is relatively small for pharmacokinetic and histological comparisons, especially given the large interindividual variability observed (e.g., paeonol Cmax at high dose). The authors should either increase the sample size, provide power calculations, or at minimum show individual concentration–time profiles to illustrate variability.

We appreciate your concern. We agree that using five animals per group may limited in several PK endpoints. We have now included the individual concentration-time profiles for paeonol and paeoniflorin in the Supplementary figure (Fig. S2 and S3) and referred them in the text (Line 242 and 273-274). Additionally, we inserted a new paragraph in the manuscript (Line 426-432) to describe this inclusion.

3) Please provide raw PK data (machine-readable) as mentioned. Reference them clearly in the main text.

We appreciate the reviewer’s comment. The raw plasma concentration-time data for all individual animals have been provided in a machine-readable format thought the Zenodo database (DOI: 10.5281/zenodo.17462573). This information has been clearly referenced in the main text (Line 163-164).

Minor

1) Given the dramatic increase in paeonol systemic exposure at the high dose, safety/toxicity monitoring is important. The manuscript currently states that no euthanasia endpoints were met, but it does not report biochemical markers (e.g., liver/kidney function) or histopathological observations beyond the colon. These should be included or at least discussed as a limitation.

We agree with your suggestion. We have expanded the discussion of limitations to acknowledge that safety biomarkers (ALT, AST, ALP,…) and multi-organ histopathology were not assessed in this study, which limits definitive conclusions on systemic toxicity (Line 433-436).

---

## [Editor Report · Decision Letter 1]

13 Nov 2025

Pharmacokinetic profiles of Moutan Cortex after single and repeated administration in a dinitrobenzene sulfonic acid-induced colitis model

PONE-D-25-45138R1

Dear Dr. Kim,

We’re pleased to inform you that your manuscript has been judged scientifically suitable for publication and will be formally accepted for publication once it meets all outstanding technical requirements.

Kind regards,

Zheng Yuan

Academic Editor

PLOS ONE
---

## [Editor Report · Acceptance letter]

PONE-D-25-45138R1

PLOS ONE

Dear Dr. Kim,

I'm pleased to inform you that your manuscript has been deemed suitable for publication in PLOS ONE. Congratulations! Your manuscript is now being handed over to our production team.

Kind regards,

on behalf of

Dr. Zheng Yuan

Academic Editor

PLOS ONE